# Fabrication of a Low-Cost Microfluidic Device for High-Throughput Drug Testing on Static and Dynamic Cancer Spheroid Culture Models

**DOI:** 10.3390/diagnostics13081394

**Published:** 2023-04-11

**Authors:** Tung Dinh Do, Uyen Thu Pham, Linh Phuong Nguyen, Trang Minh Nguyen, Cuong Nguyen Bui, Susan Oliver, Phuong Pham, Toan Quoc Tran, Bich Thi Hoang, Minh Thi Hong Pham, Dung Thuy Nguyen Pham, Duong Thanh Nguyen

**Affiliations:** 1Saint Paul General Hospital, No. 12, Chu Van An St., Ba Dinh Dist, Ha Noi 10000, Vietnam; bsdinhtung@gmail.com; 2Institute for Tropical Technology, Vietnam Academy of Science and Technology (VAST), 18 Hoang Quoc Viet St., Cau Giay Dist., Hanoi 10000, Vietnam; uyenpt.bi9231@st.usth.edu.vn (U.T.P.);; 3School of Preventive Medicine and Public Health, Hanoi Medical University, 1 Ton That Tung St., Dong Da Dist., Hanoi 10000, Vietnam; nplinh239@gmail.com; 4Hung Yen University of Technology and Education (UTEHY), 39A St., Khoai Chau Dist., Hung Yen 17000, Vietnam; cuongcdt.utehy@gmail.com; 5Centre for Advanced Macromolecular Design and Australian Centre for NanoMedicine, School of Chemical Engineering, The University of New South Wales, Sydney, NSW 2052, Australia; susan.oliver@unsw.edu.au (S.O.); thithuphuong.pham@student.unsw.edu.au (P.P.); 6Graduate University of Science and Technology, Vietnam Academy of Science and Technology (VAST), 18 Hoang Quoc Viet St., Cau Giay Dist., Hanoi 10000, Vietnamminhhcsh@gmail.com (M.T.H.P.); 7Institute of Natural Products Chemistry, Vietnam Academy of Science and Technology (VAST), 18 Hoang Quoc Viet St., Cau Giay Dist., Hanoi 10000, Vietnam; bichhoang.inpc@gmail.com; 8Institute of Applied Technology and Sustainable Development, Nguyen Tat Thanh University, Ho Chi Minh City 70000, Vietnam; 9Faculty of Environmental and Food Engineering, Nguyen Tat Thanh University, Ho Chi Minh City 70000, Vietnam

**Keywords:** cancer spheroid, drug testing, liposome, microfluidics, static and dynamic

## Abstract

Drug development is a complex and expensive process from new drug discovery to product approval. Most drug screening and testing rely on in vitro 2D cell culture models; however, they generally lack in vivo tissue microarchitecture and physiological functionality. Therefore, many researchers have used engineering methods, such as microfluidic devices, to culture 3D cells in dynamic conditions. In this study, a simple and low-cost microfluidic device was fabricated using Poly Methyl Methacrylate (PMMA), a widely available material, and the total cost of the completed device was USD 17.75. Dynamic and static cell culture examinations were applied to monitor the growth of 3D cells. α-MG-loaded GA liposomes were used as the drug to test cell viability in 3D cancer spheroids. Two cell culture conditions (i.e., static and dynamic) were also used in drug testing to simulate the effect of flow on drug cytotoxicity. Results from all assays showed that with the velocity of 0.005 mL/min, cell viability was significantly impaired to nearly 30% after 72 h in a dynamic culture. This device is expected to improve in vitro testing models, reduce and eliminate unsuitable compounds, and select more accurate combinations for in vivo testing.

## 1. Introduction

Drug development is a complex process from new drug discovery to product approval. It is an ever-evolving process that can take between 8 and 12 years, and approximately 90% of drug candidates fail at the clinical trial stage and, therefore, do not receive regulatory approval [1,2]. One reason for the high rejection rate of candidates is the inability to predict drug toxicity and efficacy in humans, owing to differences between in vitro and in vivo experiments [3,4]. Therefore, improving in vitro testing models is expected to reduce unsuitable compounds and select more accurate compounds for in vivo testing [5].

Currently, most drug screening relies on in vitro 2D cell culture models owing to the easy control of a single well-defined cell type, and they simplify the manipulation of large quantities of cells in drug screening and testing. However, these cellular models generally lack in vivo tissue microarchitecture and physiological functionality. Several previous studies on 3D cultures have shown different results for cells in morphology, viability, differentiation, and response to stimuli [6,7,8]. In an attempt to improve the success rates, developing systems that better imitate in vivo circumstances have been studied, and researchers are increasingly studying 3D cell culture platforms that allow cells to grow three directionally [7]. Cells in 3D culture platforms can mimic in vivo physiology and specific functions of actual organs, thus they have obtained considerable attention in drug testing models. Examples of this culture technology include liquid overlay culture, culture on hydrogel, bioreactor-depended culture, bio-printing, and scaffold-based [7,9]. However, these methods often have limitations, such as no support or porosity, limited flexibility, or require expensive 3D equipment and material. Furthermore, they cannot simulate the in vivo physiology and pathology of the human body.

In cell culture, the static culture is the medium supplied and replaced by a fresh medium manually at regular intervals [10]. However, in the bodies of humans and animals, the medium follows a circulating flow throughout the body, and this creates a large disparity between in vitro and in vivo studies. Therefore, dynamic systems have been researched and developed [11]. Previous studies have shown that velocity affects cell growth. For example, Jun et al.’s study demonstrated that the enhancement of health and maintenance of islet endothelial cells, and the reconstitution of the major element of islet extracellular matrix within spheroids in dynamic culture, are more prominent than in static culture [12]. Dynamic culture platforms have large mechanical forces or different physiological signals that influence the growth of cells. For example, the study by Nii et al. shows that the viability of cell aggregates under the shaking culture was significantly higher than that under static culture in the smaller ratio of microspheres [13]. In addition, Niibe’s study also applied the shaking method, which maintained and restored the “undifferentiated MSC-pool” property that was lost in the culture dish [14]. However, the limitations of this method are the size of the device or system and the high cost of the shaking machine. In addition, obtaining the mass and quick production of 3D cells for drug screening simultaneously is difficult to achieve with the existing technologies. Therefore, many researchers have used engineering methods such as microfluidic systems to solve these problems.

By using a microfluidic system that permits sufficient control of culture solutions and provides dynamic physical conditions, the spheroids are formed and cultured in large numbers while maintaining a complicated microenvironment. Microfluidic systems can be produced from a variety of materials such as silicon, glass, and polymers with diverse fabrication methods [8,15,16,17]. Due to the variety of materials and fabrication methods, different microfluidic devices have been exploited for in vitro anticancer drug testing. With the advantages introduced, microfluidic systems have attracted considerable attention for many applications in industries, such as medical, biological, and chemical. However, when put into mass production, the standard fabrication approaches have several weaknesses, such as high-cost substrates, long expensive process times, and some undesirable physical properties of conventional materials.

In this study, PMMA is used to fabricate microfluidic devices. PMMA is an amorphous thermoplastic that is widely used in microfluidic production due to its good solvent compatibility, good mechanical properties, optical transparency, and low-cost [16,18,19,20]. These properties are highly beneficial for in vitro research, especially for organ-on-a-chip devices and microfluidic systems [17]. The device is made using the laser cutting method, which is a manufacturing method that is used with diverse materials, thus allowing production in large quantities at a low cost [17]. In addition, polyethylene glycol diacrylate (PEGDA) is also proposed to be used to form microwells for tumor cultures. The benefit of PEGDA is its highly hydrophilic polymers in a cross-linked three-dimensional (3D) matrix structure, which means it can absorb large amounts of water or biological fluids. Therefore, it is capable of mimicking the natural extracellular matrix (ECM), making it a great candidate for attempting to mimic the microstructure and mechanics of an in vivo environment.

α-Mangostin (α-MG) is one of the main xanthones extracted from the rind of mangosteen. It possesses many outstanding biological activities, such as an anti-inflammatory ability [21], high antioxidant capacity [22,23], and killing cancer cells [24]. However, α-MG has a very poor water solubility (2.03 × 10^−4^ mg/L) [25]. Therefore, in treatment, as the intracellular and extracellular media are mostly water, α-MG in its natural form will have low efficiency. Secondly, the potent cytotoxic activity of α-MG can also lead to the death of normal cells [24]. Traditional drug delivery methods in chemotherapy, such as intravenous injections, cannot avoid this problem. To improve the water solubility of α-MG and reduce the death of normal cells, drug delivery systems are necessary. Liposomes are a drug delivery system that improves their efficacy and safety by delivering them to defined target sites. They comprise an aqueous core covered with lipid bilayers, which is a drug delivery system widely used to enhance a drug’s stability, efficacy, and therapeutic index [26,27]. Liposomes have some beneficial properties such as flexibility, biocompatibility, complete biodegradability, and lowered exposure of sensitive tissues [28]. To improve the precision of drug carriers toward tumors, liposomes are modified by targeting 18β-glycyrrhetinic acids (GA) ligands. According to Cai et al., the application of GA ligands in diagnosis and therapies significantly enhanced the targeting ability and treatment efficiency for Hepatocellular carcinoma cells [29]. Therefore, in this study, the GA liposome is used to deliver α-MG to protect the drug from metabolic degradation and improve the dispersion and absorption of drugs into cancer tissues.

In this study, we describe the fabrication of a low-cost and simple microfluidic device, observe the influence of velocity on cytotoxicity, and examine the effect of drug concentration on cell viability. The manufacturing, testing, and evaluation process of the device in the study are summarized in Figure 1. The device is designed and machined using laser cutting and engraving. Parts of the microfluidic chip are assembled and prepared for cell culture. Subsequently, cells are seeded in microfluidic wells and the spheroids are formed and cultured within 7 days. Then, the spheroids are examined for live/death assay and tested with α-MG-loaded GA liposomes. The results of the device evaluation suggest it could make a promising contribution to the drug development process with a new, simple, and cost-effective method for high-precision drug testing.

## 2. Materials and Methods

### 2.1. Materials

HepG2 was obtained from American Type Culture Collection (ATCC, Manassas, VA, USA). Trypsin-EDTA 0.25%, trypsin neutralizer solution, and Dulbecco’s Modified Eagle Medium (DMEM) were purchased from Gibco (Grand Island, NY, USA). Phosphate buffered saline (PBS), fetal bovine serum (FBS), sodium hydroxide (NaOH), trimethylolpropane triacrylate (TMSPMA), polyethylene glycol diacrylate (PEGDA), and Photo Initiator (PI) were bought from Sigma-Aldrich (St. Louis, MO, USA); and the Live/Dead^®^ Viability/Cytotoxicity kit from Invitrogen (Waltham, MA, USA).

A polymethyl methacrylate (PMMA) sheet with a thickness of 3 mm and 5 mm was obtained from Fusheng (Taiwan). The cutting laser fiber machine of Laser Top (Vietnam) that has a capacity of 50 W CO_2_ laser cutting machine 4060 50W (China) was used. The glasses in the sizes of 24 × 50 mm and 22 × 22 mm were purchased from Jiang Huida Medical Instruments Co., Ltd. (Yancheng, China). Polydimethylsiloxane (PDMS) prepolymer (RTV 615 A) and curing agent (RTV 615 B) were purchased from Momentive Performance Materials (Waterford, NY, USA). The red and blue dyes are food color liquids from Vinh Nam Anh Co., Ltd. (Ho Chi Minh City, Vietnam). ImageJ software was from the National Institute of Health (Bethesda, MD, USA). The OriginPro 2022 software was downloaded from OriginLab (Northampton, MA, USA).

### 2.2. Methods

#### 2.2.1. Fabrication of the Device

The device contained three layers, as shown in Figure 2B. (i) The PMMA components of the device were designed using CorelDraw 2019 software (Corel Corporation, Ottawa, ON, Canada) and manufactured with laser cutting and engraving. Before using UV glue to stack each layer, the PMMA pads were sonicated for 10 min in ethanol as a sterilizing and washing process. (ii) Two cover glasses: the top cover being square-shaped (22 × 22 mm), and the bottom cover being rectangle-shaped (24 × 50 mm). The bottom cover glass was treated to enhance the connection between the glass and hydrogel microwells using the following procedure. First, the bottom cover glass was placed in 50 mL NaOH 10% overnight. After that, the glass slides were washed with 20 mL ethanol 70% and 20 mL ethanol 100% and then dried at 70 °C ambient temperature. The glass slides were stacked on a beaker and wetted with 2 mL of TMSPMA overnight. After that, the glasses were cleaned again with distilled water, 20 mL of ethanol 70%, and 20 mL of ethanol 100%. (iii) Hydrogel microwells. The process of manufacturing this device for 3D cell culture is demonstrated in Figure 1.

The PDMS mold was fabricated using soft lithography [30] with PMMA microwell molds. A PDMS mixture with a ratio of 10:1 (RTV 615 A: B) was degassed for 30 min at 25 °C and poured onto the PMMA molds to make the microwell layer of the device. After the microwell layer was cured in an oven at 80 °C for 30 min, the layers were peeled off from the PMMA molds. The PDMS microwell mold was trimmed, cleaned, and then aligned to create hydrogel microwells. 

The microwells were formed from a hydrogel mixture that included 89% PBS, 10% PEGDA, and 1%PI. Then, 50 µL of the solution was dropped on two chambers, and the PDMS placed over them. The polymerization was initiated thermally with UV irradiation for 2 min. Finally, a square normal glass was attached after the microwells had been created to cover the top of the culture chambers.

#### 2.2.2. Leaking Test

To test for leaking in the system, a colored aqueous solution was allowed to flow through the system. The liquids with food dyes (red and blue) were connected to the inputs of the device. The flow velocity was set using a syringe pump with speeds of 0.01 mL/min, 0.05 mL/min, 0.1 mL/ min, 0.5 mL/ min, 0.8 mL/min, and 1 mL/min for the leaking test of the system with high pressure for 5 min. After 5 min, when the channels and culture chambers were covered with the dye solution, they were checked for glass breakage or leakage of dye water. The leak test was satisfactory when the two separate microchannels and culture chambers in a microfluidic chip contained only one color corresponding to the color solution in the inlet syringe. In addition, the color solution was unfurled to the other layers, and each part showed no signs of cracking.

#### 2.2.3. Cell Seeding and Cancer Spheroid Formation

To compare the effect of cell concentration, five syringes containing cells with a concentration of 10^4^, 2 × 10^4^, 5 × 10^4^, 10^5^, and 2 × 10^5^ cells/mL were used. The syringes pushed the cells into the microfluidic system directly into the microwells. After 5 min, a microscope was used to check and count the number of cells falling into the microwell. The experiment was performed in triplicate. To examine the effect of velocity on cell culture, 5 syringes containing 3 mL of solution with a cell concentration of 2 × 10^5^ cells/mL were attached to the pump with concentration settings of 0.01 mL/min, 0.03 mL/min, 0.05 mL/min, 0.1 mL/min, and 0.5 mL/min. Cells were pushed into the microfluidic system with the syringe straight into the culture area and dropped into the microwells. After 5 min, the cell microwells were examined, and the number of cells was counted under the microscope.

For cancer spheroids formation and culture in the microfluidic device, the device was placed under UV light at 37 °C for 30 min before injecting the medium. Before cell seeding, the microenvironment was cleaned with PBS. Then, the cell solution was prepared with a concentration of 2 × 10^5^ cells/mL and pumped into the inlet of the device. HepG2 cells followed the flow and went through the culture chamber to fill up the bottom of the microwells. After seeding, the microfluidic device was kept stable for 5 min for cells to deposit inside the microwell, and excess cells in the devices were removed by cleaning with DMEM medium supplemented with 10% FBS and 1% PS. The setting flow speed of the medium was 0.005 mL/min during cultivation. For static 3D cell culture, 2 mL medium was added to the inlet and outlet sides with equal solution levels after cell seeding. Cells were cultured in a DMEM medium microfluidic at 37⁰C in a 5% CO_2_ incubator for 7 days for cancer spheroid growth and viability. The DMEM cultures were refreshed after 24 h, and the formation of cancer spheroids was observed using a microscope at particular times: day 3, day 5, and day 7. Cell viability of the spheroids after 7 days of culture in the microfluidic system was determined using the live/death assay. 

#### 2.2.4. Drug Preparation and Testing

Liposome synthesis was performed using the film hydration method and followed by extrusion as described in previous reports [31]. To evaluate α-MG-encapsulated GA liposomes, assays of drug testing on the microfluidic chips were conducted. The supplemented DMEM medium containing the anticancer drug at different concentrations (0.1 mg/mL, 0.5 mg/mL, 1 mg/mL, 5 mg/mL, 10 mg/mL) was introduced into the microfluidic chambers in the single device for tumor treatment. Each device had two separate culture chambers for testing between dynamic and stationary media, as illustrated in Figure 1. In dynamic conditions, the speed set on the syringe pump was 0.005 mL/min, and the treatment was maintained for at least 24 h.

To evaluate the ability of α-MG-loaded GA liposomes to treat at variable velocities, syringes containing supplemented medium and drug at a concentration of 3 mg/mL were added to the input of the microfluidic device. The syringes were connected with the velocities set from 0.001 µL/min to 0.1 µL/min in the syringe pump, and the treatment was maintained for at least 6 h.

To further examine the influence of α-MG-loaded GA liposomes in dynamic microfluidic systems, the spheroids were treated with the drug for 60 min. After 60 min, the drug syringes in the system were replaced with syringes containing DMEM solution, and the culturing was continued for 23 h. The experiment was repeated continuously for 3 days. Live/dead assays to examine cell viability were carried out after 24 h.

#### 2.2.5. Cell Viability

Cell viability was determined using CaAM/EthD-1 staining. It is performed by incubating isolated follicles with 2 μmol/L Calcein AM and 5 μmol/L ethidium homodimer for 30 min at 37 °C in the dark. Calcein-AM is enzymatically hydrolyzed into calcein in living cells, thereby causing them to glow a bright fluorescent green color with an excitation/emission of about 495/515 nm. Ethidium homodimer-I enters cells with damaged membranes and then attaches to nucleic acids, resulting in a bright red fluorescence in dead cells with approximately 495/635 nm of excitation/emission. Fluorescence images showing the cell viability were analyzed using ImageJ software.

#### 2.2.6. Statistical Analysis

All biological experiments were performed in triplicate. Statistical analyses were performed with OriginPro 2022 with error bars showing one standard deviation. Student’s *t*-tests and ANOVA were used to determine statistical significance with *p <* 0.05.

## 3. Results and Discussion

### 3.1. Fabrication of Microfluidic Device

The microfluidic device was technically designed using CorelDraw 2019 software and visually 3D simulated using Inventor software, as shown in Figure 2A. The PMMA parts were fabricated precisely following the design. The process was quick and easy yet ensured that the parts were consistent with the design and simulation.

The first layer consisted of a 22 × 22 mm piece of glass and two pieces of PMMA of 5 mm thickness. Each piece contained two holes (Diameter (D) = 2 mm) and was attached with four tube equivalents to two inputs and two outputs of the device. The second layer was a 2 mm PMMA piece that included two input holes, two output holes (D = 2 mm), and two separated microchannel (width = 1.5 mm) systems connected to two chambers for multi-well cell culture. The bottom layer of the device was a rectangular plate of 24 × 50 mm glass. The microwell mold consisted of two pieces of 2 mm PMMA with dimensions of 22 × 22 mm, and the surface diameter of the microwells was 400 µm. 

The pieces were assembled into the final device, which is illustrated in 3D in Figure 2B, using a layer-by-layer arrangement. The glass layers had optical clarity, allowing the device to be observed clearly under the microscope [15]. The modified bottom glass surface helped the hydrogel microwells adhere more firmly to the glass and not drift following the medium flow during the cell culture and drug testing process. The formed microwells had a cone curve shape with a design surface diameter of 400 µm. In the 3D microwell culture, the shape and size of culture microwells determine the cell culture efficiency and spheroids formation [32,33]. Cubical or cylindrical microwells are commonly used in 3D cell culture. An example of the use of cubic microwells is the study of Tingjiao et al., which used microwells with the sizes of 100 µm and 500 µm [34]. However, the formation of spheres was not clearly shown. Therefore, in this study, the microwells were designed as curved microwells, which helped cells grow in and tend to form a single combination per microwell [35]. In addition, the curved microwells improved oxygen and nutrition delivery to the spheroids [32]. 

After being assembled, the device was leak-tested using red and blue water, as shown in Figure 2C. The maximum velocity that this device can withstand without causing leakage and breaking the top glass was 0.8 mL/min. Therefore, the device is capable of working comfortably at high speeds.

Currently, the materials for making microfluidic systems are diverse, but microfluidic research aimed at commercial applications has focused on creating devices using hard polymers as bulk materials [36]. PMMA is widely used in research laboratories because it is optically transparent, low-cost, and can be manipulated with various fabrication methods [20]. The prices of the components of the microfluidic device are listed in Table 1, which shows that the cost of the base materials of the system is inexpensive. The PMMA pads that are the main material in the device can be easily purchased for USD 1.05 and the total cost of the microfluidic device is USD 2.85. Therefore, the device that has been fabricated is a low-cost, easy-to-manufacture microfluidic system. In addition, the combined use of PMMA and glass plates helps the system to be chemically inert with good visibility under various microscopes [36]. This allows prototyping on a small production scale, which is very useful for research environments. In addition to PMMA, SU-8 is also a commonly used material in the laboratory. To study the application of microfluidic systems, Schuster and colleagues built an automated system to screen drugs and combine tumor organics. SU-8 3050 and SU-8 3025 were used to construct the channel layer, the chamber layer molds, the control layer, and the flow channels of the multiplexer control device [37]. However, these two materials have a market price of about USD 973.02 each [38]. The high cost of the SU-8 is a major barrier to fabricating microfluidic devices. Although the device used cheap, commercialized materials and is easy to fabricate, the use of commercially available materials limits the thickness (or height) of the PMMA sheets. The height is fixed, and it is difficult to adjust.

### 3.2. The Influence of Velocity on Cell Growth

In this study, the microfluidic device used hydrogel microwells to generate 3D cancer cells with seeding on a non-adhesive substrate, forming spheroids instead of attaching to the surface [39]. Syringes containing cells with concentrations of 10^4^, 2 × 10^4^, 5 × 10^4^, 10^5^, and 2 × 10^5^ were seeded into the microwell using the device’s microchannel system. Cells were dropped into the microwells. After 5 min, the medium was injected three times to wash excess cells from the microwell surface, ensuring that no cells grew outside the microwell during culture. The results of the experiments performed in triplicate are displayed in Figure 3A, which shows that the number of cells drifting down the microwell increased when the cell was inoculated with higher concentrations. At the lowest concentration (10^4^ cells/mL), about 42 cells descended into the microwell, and this increased to 76 cells when the concentration was 5 × 10^4^ cells/mL. At the highest concentration (2 × 10^5^ cells/mL), the microwell had over 138 cells. In the study of Khot et al., which used a concentration of 2 × 10^7^ HT29 cells/mL, an average of 377 cells were deposited into each well to generate 3D spheroids and cultured for 10 days [40]. Therefore, in this study, for 7 days, the cell concentration of 2 × 10^5^ cells/mL was selected as it had the most suitable number of cells in the microwell for spheroid formation and growth in the microfluidic platform. Moreover, all microwells were guaranteed to contain cells, indicating that this microfluidic device had the potential to produce multiple cancer spheres at the same time [41]. This could help to replace external cell spheroid methods, such as the use of extra-low attachment plates or the hanging drop method, to generate spherules off the device before being inserted into the spherules microfluidic device [42]. 

To test the effect of flow on cell seeding, syringes containing a cell concentration of 2 × 10^5^ cells/mL were attached to the syringe pump. There was a change in the number of cells after washing the excess cells. Before washing, the 0.05 mL/min velocity had the highest quantity of cells, which was nearly 70 cells per microwell. In contrast, after washing, there were about 140 cells per microwell at the same velocity. The reason for the significant difference before and after washing is that excess cells in the hydrogel surface were dropped into the microwell during the washing process, thus increasing the cell number. When cells were seeded at different velocities, the number of cells falling into the well rose with the increasing flow velocity [43]. This increase is similar to the results described by Liu et al. for the quantification of cell trapping in microwells at various flow velocities. With flow rates of 0.03 mL/min and 0.05 mL/min, which Lui et al. considered to be suitable for culturing dynamic medium 3D spheroids, the number of cells in the microwell was 63 cells and 92 cells, respectively [43]. However, when the velocity was increased to over 0.1 mL/min, the number of cells falling into the microwell decreased both before and after washing. The number of cells was 45 cells and 92 cells, respectively. At the velocity of 0.5 mL/min, the number of cells before washing was 23 cells, and it was 70 cells after washing. This shows that cells were drifting with the flow without falling into the microwells, causing the number to decrease [39].

Figure 4 shows the results after seeding HepG2 into microwells with a concentration of 2 × 10^5^ cells/mL and culturing for 7 days in both static and dynamic environments. Cell aggregates spontaneously formed inside the devices within one day (Day 1) and further grew into 3D spheroids. After 1, 3, 5, and 7 days of culture, the structure of HepG2 spheroids was confirmed, as shown in Figure 4C. The growth of the 3D cells versus time is shown in Figure 4B. The cell aggregates formed on Day 1 had an average diameter of 55.34 μm in both the static and dynamic conditions. In the static control, the spheroid sizes on Day 3, Day 5, and Day 7 were 74 μm, 141.46 μm, and 224.67 μm, respectively. Meanwhile, in the dynamic culture, the size of the cancer spheres increased sharply, and the growth rate was faster than that of the culture under static conditions. The dimensions of the spheroids measured on Day 3, Day 5 and Day 7 were 83.31 μm, 215.53 μm, and 367.24 μm, respectively. According to the study by Liu et al., HepG2 tumor spheroids in the 400 μm diameter microwell, which grew not to exceed the size of the microwell, had a diameter ranging from 330 μm to 370 μm [43]. The trend towards better growth of spheroids in dynamic environments was also shown in the study by Yun et al., where the culture of pancreatic islet spheroids in a dynamic environment led to an enhancement in the viability and function of islet spheroids [12].

In addition, Figure 4D shows the live/dead assay using CaAM/EthD-1 dual fluorescence staining in the spheroids. The total cost to implement this method for all devices is USD 19.8. The lack of red fluorescence confirms spheroid viability and demonstrates that the device is capable of successfully culturing cancer spheroids with high viability. With a dynamic culture system, cancer cells have significant growth ability and higher survival ability than with static culture [44]. Spheroids cultured in microfluidic systems larger than 150 μm in diameter will have limited diffusion for many molecules, especially O_2_ [45]. Accumulation of metabolic waste occurs when mass transport is ineffective, which leads to cell death within the culture and subsequently boosts antagonism to chemotherapy [46]. Therefore, spheroids cultured in this microfluidic device can mimic tumors in vivo.

### 3.3. The Influence of Velocity with Drug Treatment on Cell Viability

To evaluate the potential role of this microfluidic device for anticancer drug testing, α-MG-loaded GA liposomes are used as model drugs owing to their efficacy. Figure 5A shows the effect of different drug concentrations on cell viability in static and dynamic environments. After 24 h of treatment, cell viability is greatly reduced in both culture conditions at high drug concentrations. The two lowest drug concentrations were 0.1 mg/mL and 0.5 mg/mL, and the cell viability decreased to approximately 90.7% and 80.2%, respectively. The viability was 91.7% and 76.8%, respectively, in the dynamic condition, which was not significantly different from the static environment. A marked decline in static and dynamic culture conditions was shown at a drug concentration of 1 mg/mL. At this concentration, cell viability in the dynamic environment was 57%, and it was 62.5% in the static culture. When drug concentrations were further increased to 5 mg/mL and 10 mg/mL in the dynamic culture, cell viability was further decreased to 44.7% and 34.35%, respectively. Meanwhile, it only reduced to 53.17% and 44.4%, respectively, in the static culture. Furthermore, the IC_50_ value in the static environment was higher (about 8.02 mg/mL) than in the dynamic environment (about 4.46 mg/mL). The above results show that in the dynamic culture system, the cancer spheroids were more strongly affected than in the static system when treated with the drug [12,43].

As shown in Figure 5B, to further evaluate the difference and influence between the static and dynamic cultures, the 3D cancer spheroids were treated at different velocities. Syringes at the inlet of the microfluidic system containing a drug concentration of 3 mg/mL were attached to the pump at a set rate from 0.001 mL/min to 0.5 mL/min. The cancer spheroids were treated after 6 h, the flow rate was 0.001 mL/min, and the cell viability was reduced to 43.78%. When the flow rate was increased to 0.005 mL/min, the cell viability further decreased to 37.2%. However, at a flow of 0.01 mL/min, it increased to 45% and then increased further to over 50% at the velocity of 0.5 mL/min. This result indicated that a velocity from 0.001 mL/min to 0.005 mL/min was an appropriate velocity to use for the next drug test.

With the drug concentration of 5 mg/mL and the velocity of 0.005 mL/min, the results in Figure 5C highlight the difference when the cells are treated under both static and dynamic conditions. The testing timeline is summarized in Figure 5C. In the experiment, which was designed to simulate drug testing in vivo, the cancer spheroids were treated with a drug concentration of 3 mg/mL for 1 h, after which the syringe containing the drug was replaced with the culture solution, and the cancer spheroids were kept in culture for 23 h. The process was repeated two more times until the total treatment and subsequent cell culture time was 72 h from each drug treatment, and then a live/dead assay was performed to check the cell viability in the two cultures. After the first 24 h, cell viability in the static culture decreased to 67.8% while in the dynamic conditions, the viability was lower (about 57.3%). The difference between the two cultures continued to be shown after 48 h of culture, with cell viability under static conditions at 54.8% and under dynamic conditions at 42.7%. However, in the dynamic culture microfluidic system, the cell viability was only 31.0%, which was significantly lower than in the static culture system.

This test was performed to simulate drug effects on cell viability based on drug treatment in vivo assays. The results of the test show a clear difference between the static and dynamic environments under two different testing regimes. A decrease in the effect of the drug on cells in the dynamic cultures compared with static cultures is reported in Figure 5C. This shows that liquid flow affects the amount of drug adsorbed or infiltrated into cells in the dynamic system compared with static conditions [42]. This difference may be explained by the velocity possibly helping the α-MG-loaded GA liposomes bind to the cancer spheroids.

## 4. Conclusions

In this study, a low-cost microfluidic device for 3D cell culture and drug testing was fabricated and developed using a simple and easy method. Commercialized materials (e.g., PMMA and glass) are accessible and have good visual acuity with a total cost of only USD 17.75. The device consisted of two layers of glass and two layers of transparent PMMA with two separate culture chambers containing microwells. HepG2 was seeded into the microwells through the inlets of the device and subsequently formed into 3D spheroids. Both cell concentration and velocity were investigated and evaluated to determine their influence on the ability to seed cells into the system. While with velocity, the number of cells falling into the well was reduced owing to some of the cells drifting with the flow, at high concentrations, the number of cells increased significantly. Therefore, the highest concentration was selected for seeding and spheroid formation. The spheroids were cultured under both static and dynamic conditions, and it was shown that in dynamic cultures, the spheroids grew at a faster rate and with a higher survival rate. These spheroids were treated with α-MG-loaded GA liposomes using intermittent or continuous perfusion. The intermittent methods simulated physiological conditions based on drug treatment using in vivo assays. The results show the system has significant potential to study drug effects on cells in vivo. Therefore, the microfluidic platform presented here proposes a convenient and efficient device for anti-cancer drug screening and testing in biological labs using low-cost materials and a simple fabricating method.

## Figures and Tables

**Figure 1 diagnostics-13-01394-f001:**
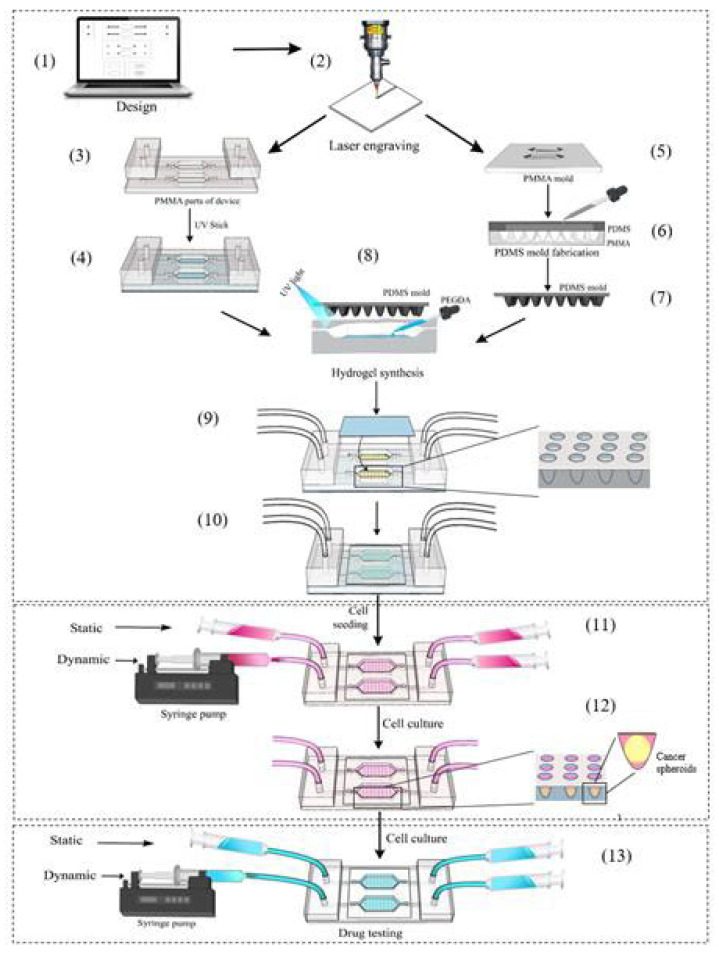
The fabrication of the microfluidic device with all sequential steps. (**1**) The PMMA components of the microfluidic device and the PMMA mold of the microwell were designed. (**2**) The PMMA sheets were laser cut and engraved according to the design. (**3**) The PMMA layers of the device were machined, and (**4**) they were attached with a UV stick. (**5**) The PMMA mold was formed. (**6**) The polydimethylsiloxane (PDMS) solution was dropped into the PMMA mold to form the PDMS mold. (**7**) The PDMS mold was generated. (**8**) The PDMS mold was placed on top of the culture area containing the hydrogel mixture of the microfluidic platform. (**9**) Microwells were generated, and a 22 × 22 mm glass plate was mounted on top of the culture area. (**10**) The microfluidic device was completed and prepared for cell seeding. (**11**) Cells were seeded into the well and then cultured in an incubator. (**12**) During the culture, cells aggregated and formed spheroids in the microwell. (**13**) After successful culture, the spheroids were used for drug testing.

**Figure 2 diagnostics-13-01394-f002:**
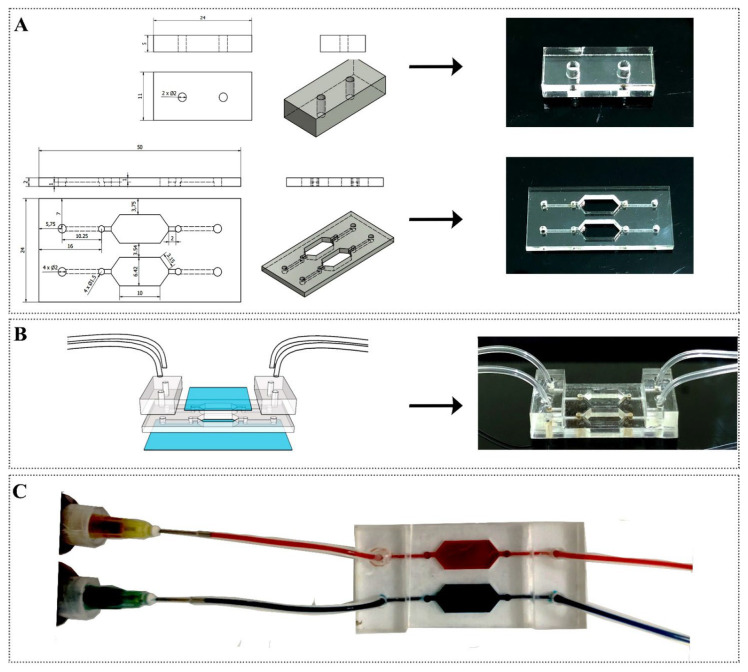
Fabrication of the microfluidic device. (**A**) The PMMA pads. (**B**) The microfluidic device is illustrated in 3D with a layer-by-layer arrangement. (**C**) The leaking test of microfluidic device with color dye and velocity of 0.8 mL/min.

**Figure 3 diagnostics-13-01394-f003:**
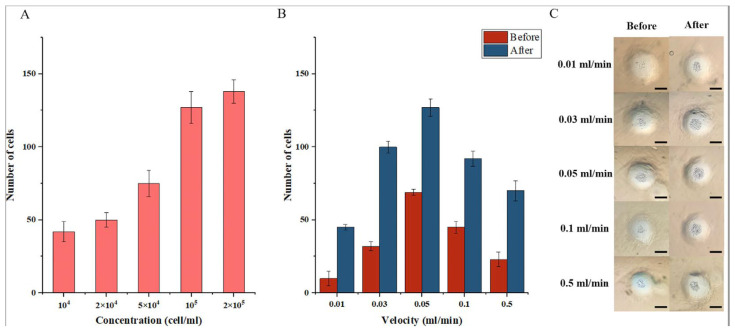
The effect of cell concentration and medium flow on cell seeding in the microfluidic device. (**A**) The number of cells seeding into the microwell increased in a cell concentration-dependent manner. (**B**) The number of cells seeding into the microwell decreased as the velocity increased from 0.1 mL/min to 0.5 mL/min. (**C**) The results are compared before and after washing and show that the excess cells on the microwell surface drifted down and increased the number of cells in the microwell (Scale bar = 150 µm, *p* < 0.05).

**Figure 4 diagnostics-13-01394-f004:**
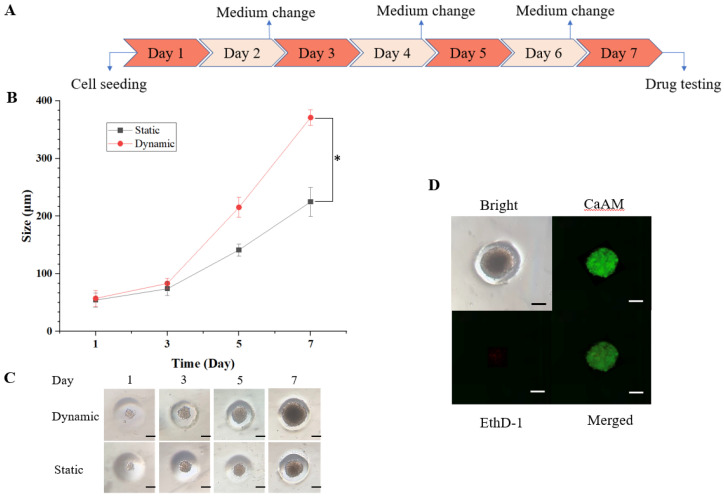
The development of cancer spheroids in the microfluidic device over 7 days. (**A**) The spheroid culture process for 7 days is summarized in this time chart. (**B**) The spheroids cultured in both static and dynamic media showed better growth under dynamic conditions, (* *p* < 0.05). (**C**) The spheroids were cultured and grown for 7 days. Scale bar = 150 µm. (**D**) The live/dead assay for the cell spheres after 7 days showed positive viability in the dynamic environment: bright image, live cells with green fluorescence (CaAM), dead cells with red fluorescence (EthD-1), and merged images.

**Figure 5 diagnostics-13-01394-f005:**
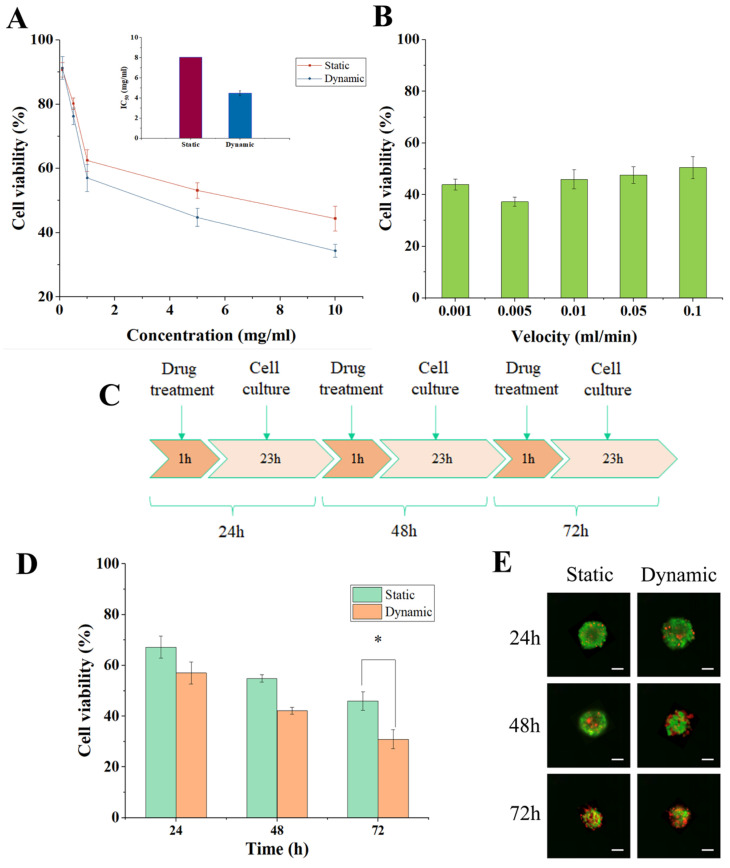
The drug testing on cancer spheroids in the microfluidic device using continuous and intermittent methods. (**A**) Test for concentrations under both static and dynamic culture conditions. The lowest cell viability is 10% at the highest drug concentration of 10 µg/mL in dynamic conditions, while cell viability under static conditions was about 27%. (**B**) Spheroids were treated with a-MG loading GA liposomes for 3 h at different velocities and cell viability was strongly attenuated to 50% and 30% at high velocities of 0.1 mL/min and 0.5 mL/min, respectively. (**C**) Intermittent drug testing diagram. (**D**) Intermittent drug testing showed that the cancer spheroids after 72 h were severely degraded to 30% in static conditions. While, in dynamic conditions, cell viability did not remarkably change. It remained at over 90% after the first 24 h of treatment and decreased slightly to nearly 80% after 72 h of treatment. (**E**) Spheroids were also stained with CaAM (Green) and EthD-1 (Red) fluorescent dyes using flow and fluorescently imaged. (Scale bar = 150 µm). Images were performed in at least 3 separate experiments (* *p* < 0.05).

**Table 1 diagnostics-13-01394-t001:** Price list for each component of the device and the total value of the microfluidic system. The price list does not include the processing fee.

	Product Name	Price (USD)	Number of Products	Amount (USD)
1	PMMA sheet	0.35	3	1.05
2	Medical tube	0.1	4	0.40
3	UV Glue 5 mL	0.5	1	0.50
4	Glass	0.05	2	0.10
5	Syringe 5 mL	0.2	4	0.80
Total				USD 2.85

## Data Availability

Data sharing is not applicable to this article as no datasets were generated or analyzed during the current study.

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
