# Peer review of "Fabrication of a Low-Cost Microfluidic Device for High-Throughput Drug Testing on Static and Dynamic Cancer Spheroid Culture Models"

_diagnostics, 2023, doi:10.3390/diagnostics13081394_

Round 1
Reviewer 1 Report
The present manuscript entitled "Fabrication of low – cost microfluidic device for high throughput drug testing on static and dynamic cancer spheroid culture models" by Do Dinh Tung, Uyen Thu Pham, Linh Phuong Nguyen, Susan Oliver, Phuong Pham, Toan Quoc Tran, Bich Thi Hoang, Minh Thi Hong Pham, Duong Thanh Nguyen, and Dung Thuy Nguyen Pham (diagnostics-2287763) is written correctly and has a good structure; moreover, it has all the necessary parts. The article is interesting from microsystems for medical application point of view; therefore, it should interest the reader. I proposed improvements in the method description and with a presentation of figures. The paper meets Diagnostics' requirements, and I recommend the article for publication in Diagnostics following the common editing stage. My current decision is a minor revision. More specific comments and observations are presented below.
1. There are a lot of double spaces in the article. They should be removed.
2. Figures. Some figures are too small to see what they represent properly. Text in drawings can be enlarged. The captions D and E are missing in the last drawing.
3. Units. Units are sometimes written with the value and sometimes separately. A liter is written as 'l' or 'L'. This should be standardised. 'u' should be replaced with 'µ' in a few cases.
4. What is the reproducibility of results on at least several prepared systems? Were other materials considered for building microdevices?
5. Page 8. I get the impression that reference [31] before [32] is missing.
6. Description of Figure 3. A and B in the description are not the same as shown in the figure.
7. Conclusions. Please clearly underline the most important advantage of the conducted research.
8. A discussion of the shortcomings and limitations of the research conducted should be added.
9. Appropriate tools should be used to best characterize the method (e.g., RGB model). The RGB model is particularly interesting here, as it can also be related to the cost of producing microsystems.
I hope that the comments presented will help improve the article.
Author Response
Thank you for your valuable comments. We have revised the manuscript and provide answers to your comments below.
- There are a lot of double spaces in the article. They should be removed.
Answer: Double spaces mistakes were removed.
- Figures. Some figures are too small to see what they represent properly. Text in drawings can be enlarged. The captions D and E are missing in the last drawing.
Answer: The caption D and E was added in the drawing. The text in figure was scaled down when edited to the magazine format. However, the images can be re-send in the sub-file that have a full size of all figure so that it can be easily followed.
- Units. Units are sometimes written with the value and sometimes separately. A liter is written as 'l' or 'L'. This should be standardised. 'u' should be replaced with 'µ' in a few cases.
Answer: The units have “l” and “u” were standardised.
- What is the reproducibility of results on at least several prepared systems? Were other materials considered for building microdevices?
Answer: The device is manufactured in large quantities and reused many times with relatively similar results. Other materials also considered for building microdevice in the future study, however, the materials must be biocompatible, low-cost and simple to fabricate.
- Page 8. I get the impression that reference [31] before [32] is missing.
Answer: The reference [31] was changed, added and highlighted in the manuscript and ref list
- Description of Figure 3. A and B in the description are not the same as shown in the figure.
Answer: The description had a reversed mistake. So, it was swapped and highlighted in the manuscript.
- Conclusions. Please clearly underline the most important advantage of the conducted research.
Answer: The conclusion was re-written to emphasize the device's advantages.
- A discussion of the shortcomings and limitations of the research conducted should be added.
Answer: The short paragraph of limitations of the research was added and marked in the manuscript.
- Appropriate tools should be used to best characterize the method (e.g., RGB model). The RGB model is particularly interesting here, as it can also be related to the cost of producing microsystems.
Answer: The cost of using RGB system was added and highlighted in the Table 1.
Reviewer 2 Report
The novelty is not clear. There are many papers on the dynamic culture of spheroids. The description was not written, and the authors should discuss the novelty by comparing these related papers. In addition, the statistical analysis was not performed, so it is impossible to evaluate the data. Taken together, major revision (nearly rejected) should be made before re-submission.
1.
As mentioned above, the authors must introduce the research on the dynamic culture of spheroids in the introduction and compare it with these papers by quoting these papers.
https://doi.org/10.1016/j.jbiosc.2019.04.013
https://doi.org/10.3389/fbioe.2020.590332
Lab Chip, 2010, 10, 1671-1677
2.
All data should be evaluated after the statistical analysis.
Author Response
Thank you for your valuable comments. We have revised the manuscript and provide answers to your comments below.
1.As mentioned above, the authors must introduce the research on the dynamic culture of spheroids in the introduction and compare it with these papers by quoting these papers.
https://doi.org/10.1016/j.jbiosc.2019.04.013
https://doi.org/10.3389/fbioe.2020.590332
Lab Chip, 2010, 10, 1671-1677
Answer: The T-test and ANOVA test are 2 methods was applied to performer. And dynamic culture of spheroids in was introduced with more data and compared with the first two papers. The last paper also added in the discussion for easy compared with our study. The novelty of this study is focus on the low-cost and quick preparation to produce a large number effective device.
- All data should be evaluated after the statistical analysis.
Answer: Thank you for your comment. We have provided the Statistical analysis section in the revised manuscript.
Reviewer 3 Report
This is a report of an inexpensive and easy-to-produce device for drug testing in cell spheroids. The manuscript is generally well-written, with a clear structure and golden thread. The introduction reads very well and the methods section contains sufficient detail. However, explanations for abbreviations like PMMA or PEGDA should be provided at first mention. What is also missing is a discussion of the potential limitations and disadvantages of this method.
An error occurred in the Figure 3 caption: The captions for A and B have been swapped.
Author Response
Thank you for your valuable comment. We have revised the manuscript and provide answer to your comment below.
An error occurred in the Figure 3 caption: The captions for A and B have been swapped.
Answer: The explanations for PMMA and PEGDA were marked in the manuscript. The caption of Figure A and B were fixed. And the the potential limitations of this method were written and highlighted in the manuscript.